# Challenges in clinical diagnosis of Clade I Mpox: Highlighting the need for enhanced diagnostic approaches

**Josephine Bourner**[1]*, **Esteban Garcia-Gallo**[1], **Festus Mbrenga**[2], **Yap Boum, II**[2], **Emmanuel Nakouné**[2], **Amy Paterson**[1], **Benjamin Jones**[1], **Piero Olliaro**[1], **Amanda Rojek**[1]

**1** ISARIC, Pandemic Sciences Institute, University of Oxford, Oxford, United Kingdom, **2** Institut Pasteur de Bangui, Bangui, Central African Republic

* josephine.bourner@ndm.ox.ac.uk

**Data Availability Statement:** The full dataset used in the study is provided in S3 Appendix.

**Funding:** This work was supported by the UK Foreign, Commonwealth and Development Office

## Abstract

### Background

Due to limited diagnostic capacity and availability of point-of-care tests, diagnosis of Clade I mpox in the geographical regions most affected is usually on clinical grounds. This may be complicated due to the similarity between mpox and varicella (chickenpox) lesions. Visual assessment of lesions is also used for determining clinical progress and to assess patient outcomes in clinical trials. However, there has been no investigation into whether clinicians can (i) identify Clade I mpox compared to other viral lesions (ii) differentiate between Clade I mpox lesion stages.

### Methodology/Principle findings

The objective of this study was to evaluate inter-rater reliability and agreement between clinicians assessing lesions in patients with Clade I mpox. We presented experienced clinicians with 17 images of Clade I mpox or varicella and asked them to independently indicate the most likely diagnosis–mpox or varicella–and to categorise the lesions according to their stage. When selecting the most likely diagnosis, accuracy varied across all images, the inter-rater reliability was poor ($\kappa = 0.223$; $z = 10.1$) and agreement was moderate ($P_o = 68\%$). When categorising lesions according to their type, if a single lesion type was present in the image, inter-rater reliability was moderate ($\kappa = 0.671$, $z = 40.6$) and agreement was good ($P_o = 78\%$), but when multiple lesion types were shown in an image, both inter-rater reliability ($\kappa = 0.153$, $z = 10.5$) and agreement ($P_o = 29\%$) decreased substantially.

### Conclusions

This study demonstrates that there are presently limitations in using visual assessment to diagnose Clade I mpox and evaluate lesion stage and treatment outcomes, which have an impact on clinical practice, public health and clinical trials. More robust indicators and tools are required to inform clinical, public-health, and research priorities, but these must be implementable in countries affected by mpox.

and Wellcome [215091/Z/18/Z] and the Bill & Melinda Gates Foundation [OPP1209135]. For the purpose of Open Access, the author has applied a CC BY public copyright licence to any Author Accepted Manuscript version arising from this submission. The funders had no role in study design, data collection and analysis, decision to publish, or preparation of the manuscript.

**Competing interests:** The authors have declared that no competing interests exist.

## Author summary

Mpox is a zoonotic illness caused by the monkeypox virus (MPXV), for which there are two distinct sub-clades. Clade I is typically found in central Africa and is associated with worse patient outcomes than Clade II. Diagnosis of mpox is most commonly performed using PCR, but in settings with limited laboratory capacity diagnosis is usually performed on clinical grounds taking lesion presentation in to account. Lesion presentation is also used to assess patient outcomes in both clinical and research settings. However, there has been no investigation into whether clinicians can (i) identify Clade I mpox compared to other viral lesions (ii) differentiate between Clade I mpox lesion stages, which has important implications for clinical practice, research and public health. Our study, which presented 16 clinicians with 17 sets of images of Clade I mpox or varicella and asked them to i) provide the most likely diagnosis and ii) categorise the lesions in to their stages, demonstrates that there are presently limitations in using visual assessment to diagnose Clade I mpox and evaluate lesion stage and treatment outcomes. Alternative methods and tools are therefore required that can be easily implemented in affected countries.

## Introduction

Mpox is a zoonotic illness caused by the monkeypox virus (MPXV), for which there are two distinct genetic sub-types, referred to as Clades. [1] Clade I mpox is found primarily in central Africa and is associated with worse patient outcomes than Clade II, which is further divided into Clade IIa (historically reported in west Africa) and Clade IIb, which caused a Public Health Emergency of International Concern (PHEIC) in 2022. [1]

There is a growing epidemic of Clade I mpox in the Democratic Republic of the Congo (DRC)–with a doubling in the number of cases over last year's. [2] There are several concerning features to this outbreak, including an expansion in geographical areas affected, new introduction to dense urban populations (including Kinshasa, the capital city of DRC), and the first descriptions of sexual transmission of this clade. Both the World Health Organization (WHO) and European Centre for Disease Prevention and Control risk assessments (ECDC) highlight that improving awareness and support for clinicians to diagnose cases is a key response priority. [2,3] While Clade II mpox is rarely fatal– 156 deaths out of over 91,000 cases reported outside Africa since May 2002 –, mortality associated with Clade I mpox is 1–12% [1] and there have been 581 deaths reported out of over 12,500 cases in DRC since February 2023. [2]

However, diagnosis of mpox is difficult. Confirmation of mpox diagnosis is primarily by PCR, [4,5] but, concerningly, new evidence suggests that the Clade I-specific RT-PCR test recommended by the US CDC is impacted due to genetic mutations in the virus causing this outbreak. [6] There are no point-of-care or rapid diagnostic tests alternatives at present. Laboratory diagnosis is impacted by operational challenges and limited laboratory capacity–in the present DRC epidemic, only 9% of suspected mpox cases have been tested by PCR. [7]

Therefore, diagnosis of Clade I mpox in the regions most affected is usually on clinical grounds. While clinical diagnosis is multifaceted and takes into consideration factors such as patient demographics, prevailing epidemiology, and clinical history–one important component is visual assessment of skin lesions. This is reported as difficult due to the similarity between mpox and varicella (chickenpox) lesions. Visual assessment of lesions is also used for determining clinical progress. Whether a lesion is 'active', 'inactive' or 'resolved' is used for determination of infectivity and infection-prevention and control requirements, decisions around whether to offer a patient access to potential treatments, and declarations of cure. The

assessment of lesions has in addition become the focal area of evaluation in clinical trials. Several trials for Clade I and Clade II mpox use time to lesion resolution as the primary endpoint to indicate treatment success. [8–12]

Despite the myriad uses of clinical diagnosis of Clade I mpox, there has been no investigation into whether clinicians can (i) identify Clade I mpox compared to other viral lesions (ii) differentiate between Clade I mpox lesion stages–although some suspected limitations of this endpoint have already been identified. [13] A previous exercise undertaken by these authors on Clade IIb lesions demonstrated only moderate agreement among different assessors. [14]

This study evaluates agreement between clinicians on a differential diagnosis between Clade I mpox and varicella, and classification of lesion stages. Whether mpox Clade I lesions can be reliably classified on clinical grounds has important implications both in the low-resource settings where the virus circulates and in the event of spread to historically non-endemic regions–for decisions on treatment, public health control, and for the robustness of clinical research.

## Methods

### Ethics statement

The objective of this study was to evaluate inter-rater reliability and agreement between clinicians assessing lesions in patients with Clade I mpox. Two focal areas of agreement and reliability were evaluated: 1) differential diagnosis between Clade I mpox and varicella based on lesion presentation; 2) categorisation of lesion stages. The design, conduct and results of this study are reported according to the Guidelines for Reporting Reliability and Agreement Studies (GRAAS). [15] Ethical approval for this study was obtained from the University of Oxford Medical Science Interdivisional Research Ethics Committee (R84355/RE001). All potential participants were sent an information sheet describing the purpose of the study and how their data would be used. Before accessing the questionnaire, participants were asked to declare that they had read the information sheet and give written informed consent to participate.

### Participants

The participants in this study were clinicians who had experience treating and managing patients with Clade I mpox. Due to the potentially limited pool of participants with appropriate experience in the management of Clade I mpox, no target sample size was defined in advance of conducting this study. The study was therefore designed to be descriptive.

In total, 38 clinicians were contacted to participate: 15 from the DRC, seven from the Central African Republic, seven from France, three from Belgium, three from Switzerland, two from the UK and one from Nigeria (**Fig 1**).

All potential participants were sent an information sheet describing the purpose of the study and how their data would be used. Before accessing the questionnaire, participants were asked to declare that they had read the information sheet and give written informed consent to participate.

### Data collection

Using the RedCap survey tool [16,17], participants were shown 17 sets of images of patients with either Clade I mpox or varicella lesions confirmed by PCR. The anonymised images selected for use in the survey were derived from the Institut Pasteur de Bangui clinical image library and showed lesions of different stages and located on different areas of the body.

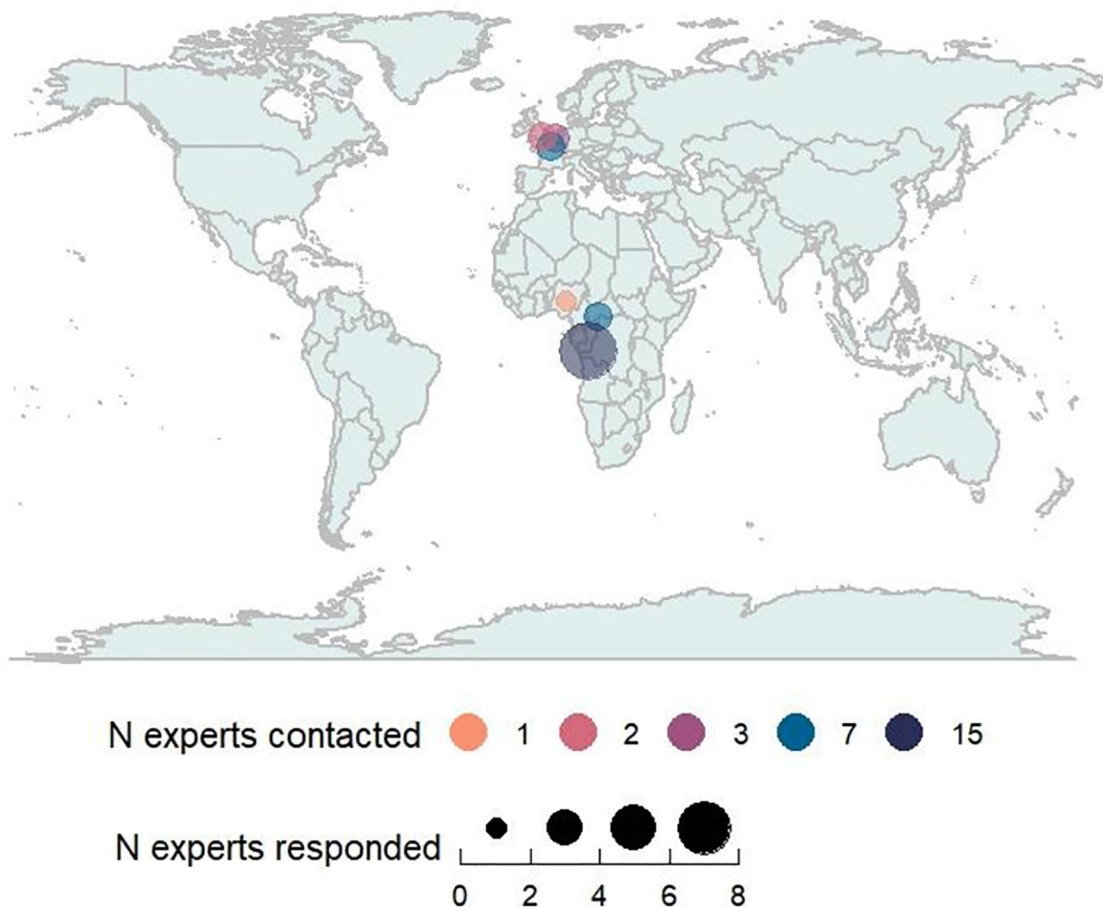

**Fig 1. Number of expert clinicians contacted and number of expert clinicians who returned a response by country.** The base map in Fig 1 was generated using the 'ggplot2' package in R.

Based on the lesion presentation in the images, the participants were asked to independently indicate the most likely diagnosis–mpox or varicella–and to categorise the lesions as being either active, scabbed or resolved, or they had the option to state they were unable to classify the lesions. Before starting the questionnaire, participants were shown the World Health Organisation's working definition of each lesion stage. [18]

The survey also captured each clinician's country of practice, average number of patients they manage with Clade I mpox in a year, and their self-rated confidence evaluating mpox lesions. Confidence evaluating mpox lesions was assessed on an ordinal scale from one to ten, with one representing no confidence and ten representing complete confidence.

The questionnaire used in this study can be found in **S1 Appendix** and was provided in both English and French.

## Data analysis

All analyses completed in this study were conducted with the 'irr' package in R Statistical Software (v4.3.2) [19] and validated using 'statsmodels' package in Python (v3.11.5) by two analysts. The base map in Fig 1 was generated using the 'ggplot2' package and 'map_data('world')' function in R. [19]

Accuracy identifying mpox or varicella was evaluated using the percentage of raters selecting the correct diagnosis for each image. Spearman's rank correlation coefficient (Spearman's ρ) was employed to assess the relationship between experience—assessed by self-reported expertise and the number of patients managed—and each evaluator's accuracy percentage. Spearman's method, chosen for its suitability with small samples and ordinal data without assuming a normal distribution, necessitates a cautious interpretation of results due to the reduced statistical power and potential influence of noise and outliers inherent to limited datasets.

Inter-rater reliability was evaluated using Fleiss' kappa coefficient (κ) and inter-rater agreement was evaluated using proportion of partial agreement ($P_o$), and proportion of exact agreement ($P_{oe}$). Partial agreement refers to the agreement between any one of the participants' selections in a multiple-choice answer and exact agreement refers to a complete match between participants' selections in a multiple-choice answer.

Questions related to lesion classification assessed only the images in which the patient had a confirmed diagnosis of mpox. The first analysis evaluates responses to the images where only a single or homogenous lesions were present and for which the κ and $P_{oe}$ are reported. The second analysis evaluates responses to the images where multiple types of lesions (combinations of active, scabbed and resolved) are present, for which the κ, the $P_o$ the $P_{oe}$ are reported.

## Results

The questionnaire was sent to 38 potential participants in total, of whom 17 accessed the questionnaire and completed the eligibility check. One respondent was not eligible to participate as they were not directly involved in the clinical management of patients with Clade I mpox.

All 16 participants who started the questionnaire completed it in full. A summary of the participants' country of work, confidence assessing an mpox lesion and number of mpox patients they have personally managed is summarised in **Table 1** and **Fig 1**.

### Diagnosis

Accuracy selecting the most likely diagnosis between mpox and varicella based on lesion presentation varied across all images, with between 25% and 100% of the respondents correctly

**Table 1. Description of study participants.**

| | |
|---|---|
| **Country of work, n (%):** | |
| Democratic Republic of Congo | 8 (50%) |
| Central African Republic | 2 (13%) |
| France | 2 (13%) |
| United Kingdom | 2 (13%) |
| Nigeria | 1 (6%) |
| Belgium | 1(6%) |
| **Number of mpox patients the respondents has personally managed, n (%):** | |
| <5 | 0 |
| 5–10 | 2 (13%) |
| 10–20 | 1 (6%) |
| 20–50 | 5 (31%) |
| >50 | 8 (50%) |
| **Confidence assessing mpox lesions, median (IQR):** | |
| Confidence score | 8 (7.5–8) |

**Table 2. Correlation between raters' diagnostic accuracy, self-rated confidence and experience using Spearman's correlation (ρ).**

|  | Image | Self-rated confidence | Experience |
|---|---|---|---|
| Image | - | 0.3 | 0.168 |
| Self-rated confidence | 0.3 | - | -0.042 |
| Experience | 0.168 | -0.042 | - |

identifying the disease and a median = 75% (Q1: 62.5%, Q3: 93.75%). No correlation was detected between accuracy and either self-rated confidence (Spearman's ρ = 0.3) or experience (Spearman's ρ = 0.17) assessing lesions (**Table 2**).

When asked to select the most likely diagnosis of the patient between varicella and mpox based on the lesion presentation shown in the images, the inter-rater reliability was poor ($\kappa$ = 0.223; z = 10.1) and agreement was moderate ($P_o$ = 68%). (**Fig 2**)

In addition to the reported results, in S2 Appendix we show that there appears to be improved diagnostic accuracy if a single lesion type is present in the image, compared to images in which lesions appeared to be in multiple concurrent stages of evolution (**S2 Appendix**). We did not observe any relationship between diagnosis and accuracy (**S2 Appendix**). However, our sample size is too small to draw any generalisable conclusions.

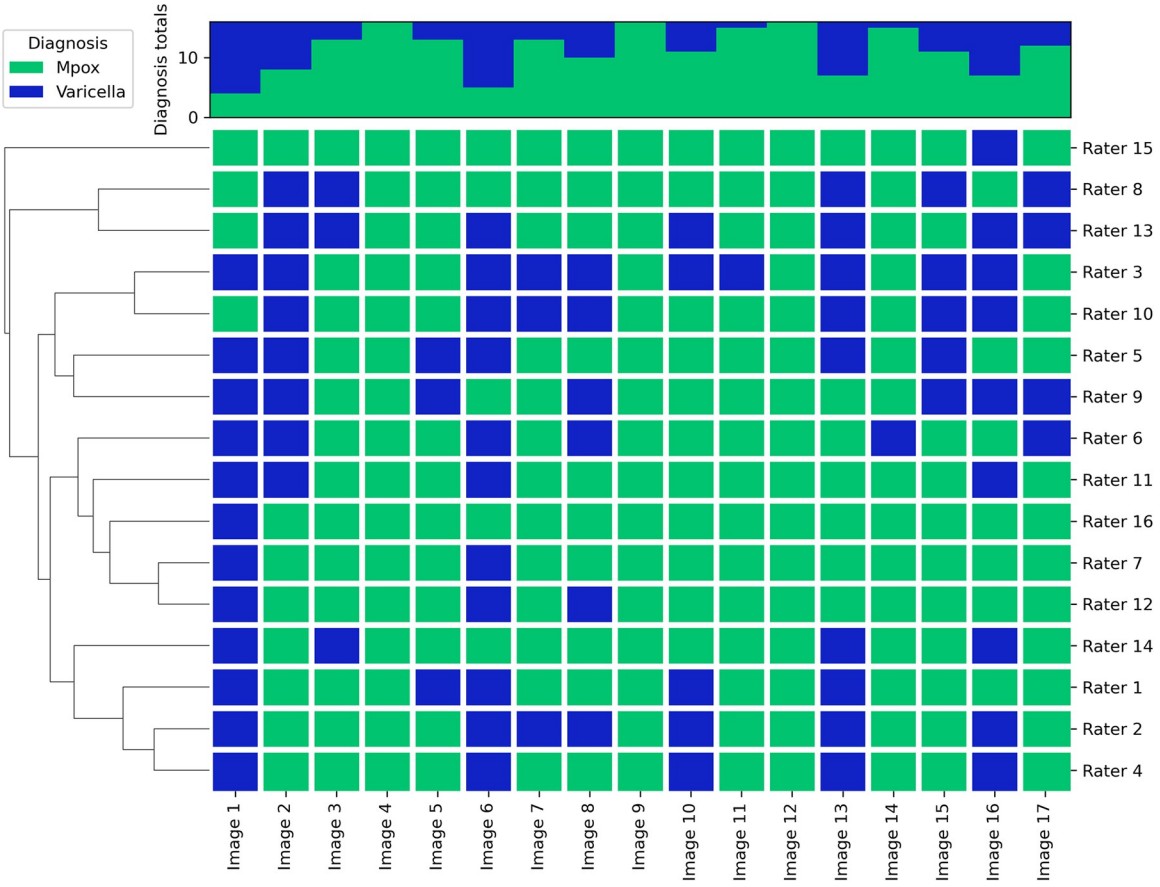

**Fig 2. Dendrogram of the suspected diagnosis associated with the images based on lesion presentation.**

**Table 3. Summary of inter-rater reliability and agreement results.**

| Diagnosis of patients in images | Response selection formats | Lesion types present in image | Fleiss' kappa | Z score | Percentage of agreement | |
|---|---|---|---|---|---|---|
| | | | | | Type | Result |
| Mpox & Varicella | Single choice | Single types and multiple types | 0.223 | 10.1 | Percentage of exact agreement | 67.745 |
| Mpox | Single choice | Single types | 0.671 | 40.6 | Percentage of exact agreement | 78.14 |
| Mpox | Multiple choice | Multiple types | 0.447 | 25.9 | Percentage of partial agreement | 73.18 |
| Mpox | Multiple choice | Multiple types | 0.153 | 10.5 | Percentage of exact agreement | 29.04 |

### Lesion classification

When a single lesion type was present in the image, inter-rater reliability was moderate ($\kappa$ = 0.671, z = 40.6) and agreement was good ($P_o$ = 78%) (**Table 3 and Fig 3**).

However, when multiple lesion types were shown in an image, both inter-rater reliability ($\kappa$ = 0.153, z = 10.5) and agreement ($P_o$ = 29%) decreased substantially. (**Table 3**)

Where partial agreement was assessed, inter-rater reliability was moderate ($\kappa$ = 0.447, z = 25.9) and agreement was good ($P_o$ = 73%). (**Fig 4** and **Table 2**)

## Discussion

This study demonstrates that there are presently limitations in using clinical assessment to diagnose Clade I mpox and evaluate lesion stage and treatment outcomes.

We found moderate accuracy, poor reliability, and moderate agreement among clinicians deciding between mpox and varicella diagnosis based on lesion presentation. In settings where

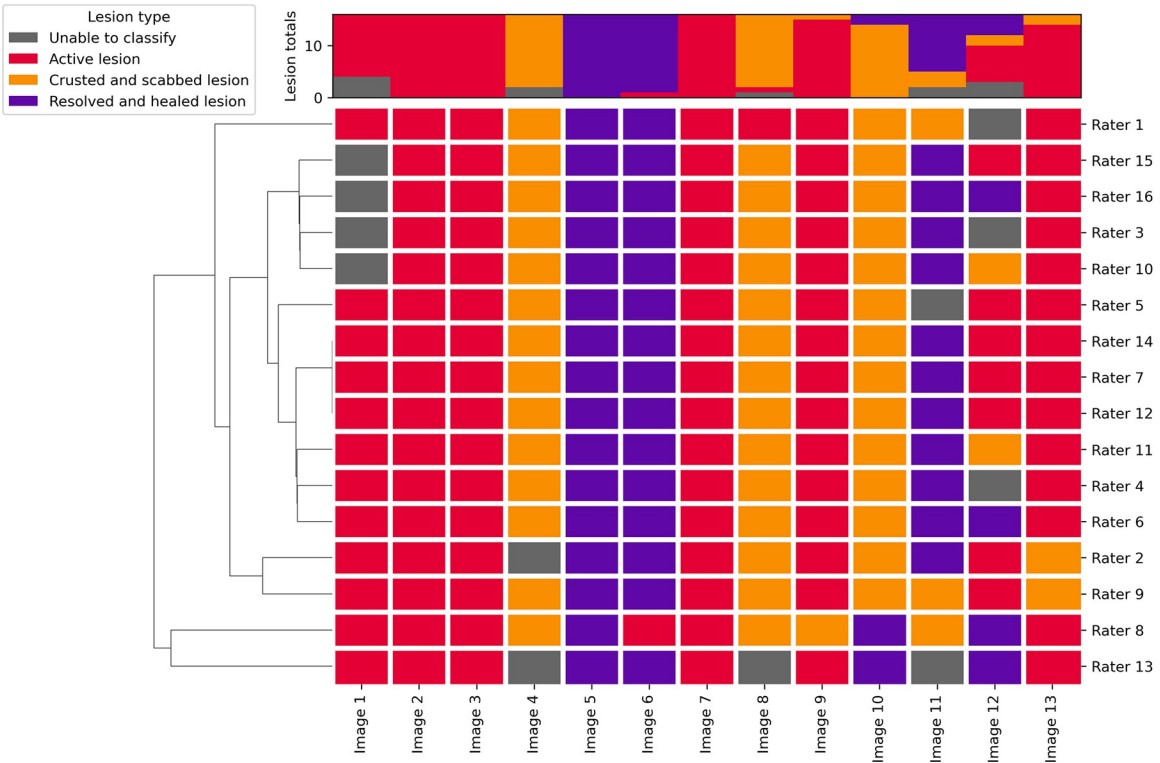

**Fig 3. Dendrogram of the respondents' lesion assessments for which a single lesion type was present in the image.**

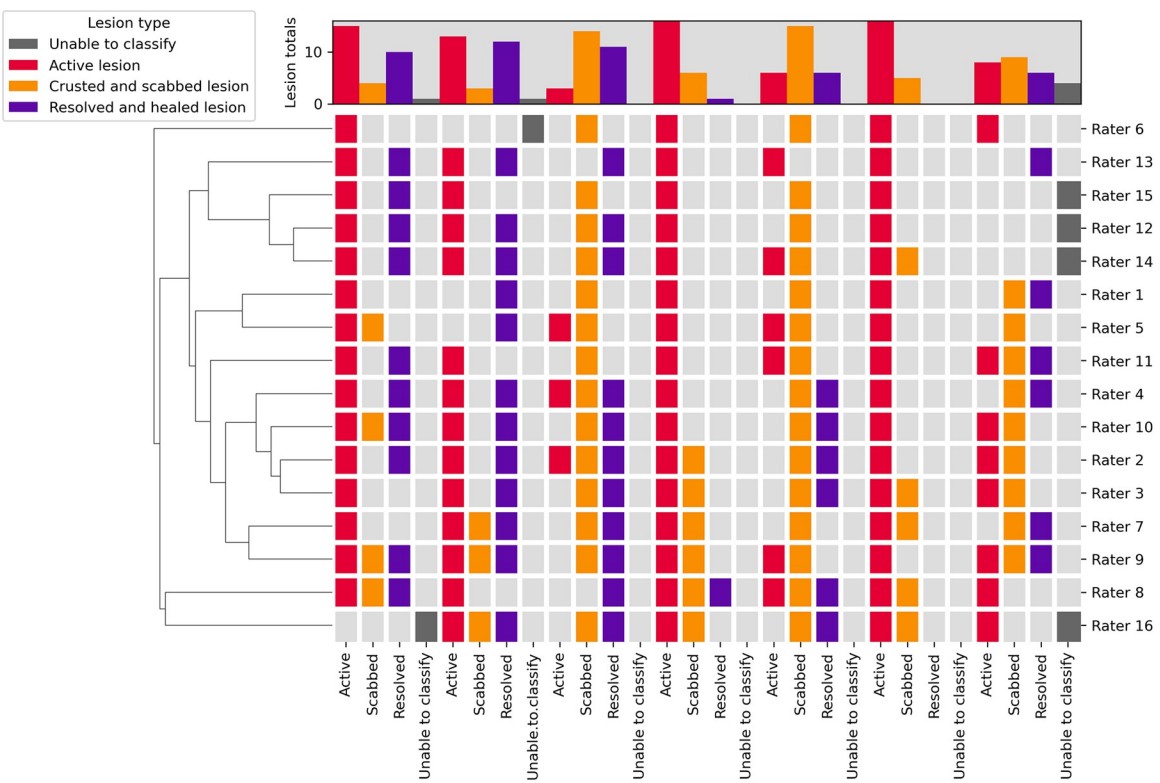

**Fig 4. Dendrogram of the respondents' lesion assessments for which multiple lesion types were present in the image.**

laboratory diagnosis is not available, or delayed, there is a risk that patients could be managed according to an incorrect care pathway (e.g. provision of an incorrect antiviral). Misclassification of cases can have implications for public health activities as varied as disease surveillance, allocation of vaccinations, drug procurement, and reporting of notifiable diseases obligations. This has consequences also for the evaluation of treatment effects, both in clinical practice and research, with incorrect or inconsistent classification of outcomes.

We found that for classification of mpox lesion status there was moderate reliability and good agreement if a single lesion type is present–although this may not be sufficient to state that there is good agreement beyond chance [20,21]–but it appears to be most challenging to obtain consistent assessments when multiple lesion types are present. As lesions are unlikely to follow the same evolution pattern over time, the poor reliability and agreement between clinicians when multiple lesion types are present creates a substantial challenge. This includes clinical trials, which rely on a single overall assessment of a patient's lesion presentation at a specific timepoint to act as an indicator of a patient's overall outcome and response to treatment.

These findings occurred despite the self-rated confidence among the participants being high (IQR: 7.5–8). We found no correlation between confidence and experience defined according to the number of patients with mpox the respondent had managed (Fig 5). Due to the sporadic and widespread reporting of Clade I mpox across a large geographic area, it is possible that confidence, and agreement among real-world clinicians, who may only occasionally manage a patient with mpox or varicella, may be lower than reported in this study where we sampled a relatively expert group.

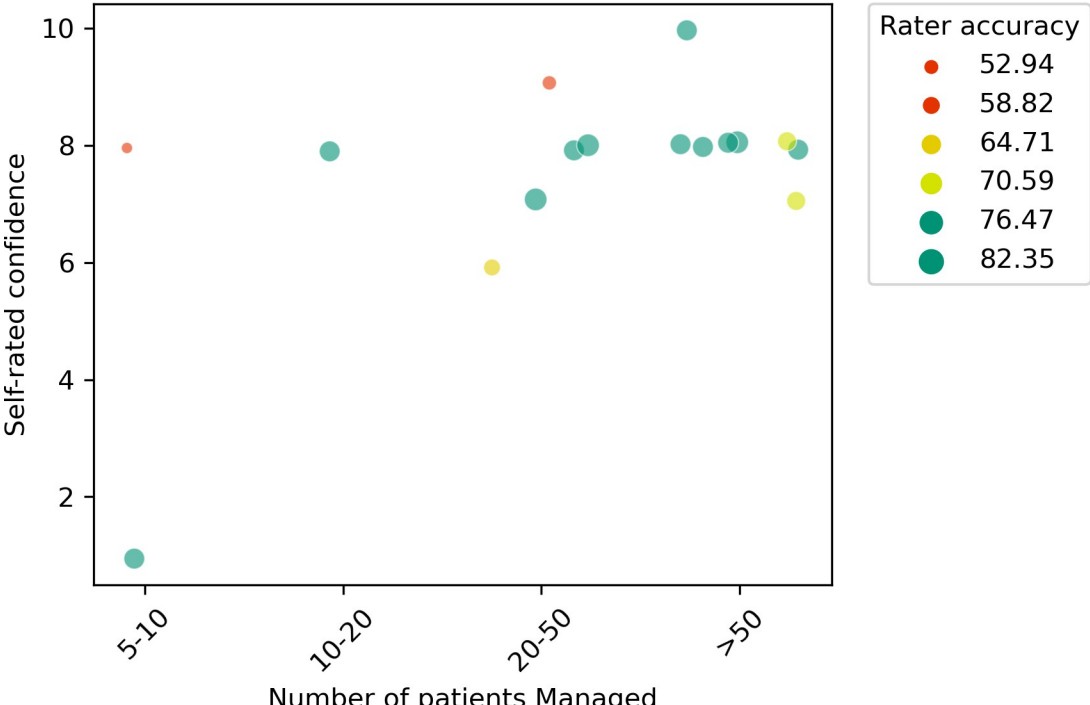

**Fig 5. Scatterplot of respondents' self-rated confidence plotted against experience (measured according to the number of patients managed).**

A parallel study found higher agreement and reliability among clinicians assessing Clade IIb mpox lesions. [14] This may be due to differences in lesion presentation between the two diseases, with Clade IIb lesions often being more localised and fewer than Clade I lesions. The surveys were not combined due to the significant differences in clinical presentation between the clades, and that clinical experience spanning both clades is limited to a very small number of highly-expert clinicians. We searched the literature for analogous studies for other epidemic-prone rash diseases, but were unable to find equivalent research. This might be due to the better understanding of the natural history of these diseases (such as presence of other diagnostic discriminators such as orchitis, or the centrifugal pattern of a rash) and also the longstanding availability of diagnostics in high-income settings.

There is a continued requirement for resources (such as the WHO lesion assessment atlas) to assist clinicians and clinical triallists working in low-resource environments to improve their assessments, alongside support for improved diagnostic capacity. There is an urgent need to work to improve and find other correlates of disease progression (e.g. how much PCR status reflects ongoing infectivity).

Considering the lack of correlation between clinician experience and their ability to diagnose Clade I Mpox, this study underscores the potential of Artificial Intelligence (AI) and Machine Learning (ML) in improving Mpox diagnosis, especially in resource-constrained settings where access to doctors may be limited and initial assessments are often conducted by nurses or community health workers. The application of AI and ML technologies, capable of capturing and analysing images, presents a significant opportunity to enhance the diagnostic accuracy of clinicians and nurses in areas lacking PCR testing facilities.

## Limitations

This study included a relatively low number of participants, because few clinicians worldwide have current or recent experience managing patients with Clade I mpox. The response rate was also challenged by difficulty identifying and reaching clinicians who work in remote areas with limited internet connection. Assessments made by visual inspection of images may not be reflective of all the information used to make assessments in a clinical scenario (such as epidemiological context, other clinical signs and symptoms of mpox such as presence of fever and adenopathy, patient's descriptions of lesion evolution, and associated characteristics such as pain). We used reliability and agreement for assessment of mpox lesions (compared to accuracy) because there is no 'gold standard' classification to which to compare. We did not provide images where there was mpox and varicella coinfection.

## Conclusion

It is difficult for experienced clinicians to distinguish clade I mpox from varicella, and to reliably assess disease stage for clade I mpox. More robust indicators and tools are required to inform clinical, public-health, and research priorities, but these must be implementable in countries affected by Mpox.

## Supporting information

**S1 Appendix. Questionnaire.**
(PDF)

**S2 Appendix. Summary of question characteristics and accuracy score.**
(PDF)

**S3 Appendix. Full dataset.**
(CSV)

## Author Contributions

**Conceptualization:** Josephine Bourner, Amy Paterson, Benjamin Jones, Piero Olliaro, Amanda Rojek.

**Data curation:** Josephine Bourner.

**Formal analysis:** Josephine Bourner, Esteban Garcia-Gallo.

**Funding acquisition:** Piero Olliaro.

**Investigation:** Festus Mbrenga, Amy Paterson, Benjamin Jones, Amanda Rojek.

**Methodology:** Josephine Bourner, Amy Paterson, Benjamin Jones, Piero Olliaro, Amanda Rojek.

**Project administration:** Josephine Bourner, Festus Mbrenga, Yap Boum, II, Emmanuel Nakouné, Amy Paterson, Benjamin Jones.

**Resources:** Yap Boum, II, Emmanuel Nakouné, Piero Olliaro, Amanda Rojek.

**Supervision:** Emmanuel Nakouné, Piero Olliaro, Amanda Rojek.

**Validation:** Esteban Garcia-Gallo.

**Visualization:** Esteban Garcia-Gallo.

**Writing – original draft:** Josephine Bourner.

**Writing – review & editing:** Josephine Bourner, Esteban Garcia-Gallo, Festus Mbrenga, Yap Boum, II, Emmanuel Nakouné, Amy Paterson, Benjamin Jones, Piero Olliaro, Amanda Rojek.

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
