## [Decision Letter · Decision Letter 0]

22 Apr 2024

Dear Ms Bourner,

Thank you very much for submitting your manuscript "Challenges in Clinical Diagnosis of Clade I Mpox: Highlighting the Need for Enhanced Diagnostic Approaches" for consideration at PLOS Neglected Tropical Diseases. As with all papers reviewed by the journal, your manuscript was reviewed by members of the editorial board and by several independent reviewers. The reviewers appreciated the attention to an important topic. Based on the reviews, we are likely to accept this manuscript for publication, providing that you modify the manuscript according to the review recommendations. 

Sincerely,

Wen-Ping Guo

Academic Editor

Michael Holbrook

Section Editor

Reviewer's Responses to Questions

**Key Review Criteria Required for Acceptance?**

**Methods**

-Are the objectives of the study clearly articulated with a clear testable hypothesis stated?

-Is the study design appropriate to address the stated objectives?

-Is the population clearly described and appropriate for the hypothesis being tested?

-Is the sample size sufficient to ensure adequate power to address the hypothesis being tested?

-Were correct statistical analysis used to support conclusions?

-Are there concerns about ethical or regulatory requirements being met?

Reviewer #1: The objectives and hypothesis are clear. The study is reasonable given the working conditions of the target of the survey. The persons to whom the survey were sent is not defined, only as mpox experts or experienced clinicians. It is important to identify more clearly to whom the survey was sent, and it would improve clarity to explain it was sent electronically to persons working remotely in central Africa (to the extent that is correct). The desired sample size was not defined. However, this is a descriptive study and the confidence limits on the resultant measures are provided. The analysis is reasonable, comparing the evaluation of pictures of skin lesions with the underlying diagnoses of mpox or varicella, and measuring agreement among observers.

Reviewer #2: (No Response)

**Results**

-Does the analysis presented match the analysis plan?

-Are the results clearly and completely presented?

-Are the figures (Tables, Images) of sufficient quality for clarity?

Reviewer #1: The analysis presented corresponds to the plan and are clearly presented. I believe some of the figures are not necessary and do not add to the presentation of the results.

Reviewer #2: (No Response)

**Conclusions**

-Are the conclusions supported by the data presented?

-Are the limitations of analysis clearly described?

-Do the authors discuss how these data can be helpful to advance our understanding of the topic under study?

-Is public health relevance addressed?

Reviewer #1: In practice, clinicians would have more data than simply the visual inspection of skin lesions to determine if a patient has mpox (disease from MPXV infection) or varicella, and to determine the stage of mpox. Clinicians would know the age of the patient, would know the status of others in the family and perhaps in the community, would know the duration of the illness, and may know something of the vaccination status of the patient. So this study does not replicate the situation of making a diagnosis in the field. At the same time, it provides information on the uncertainty of basing the diagnosis of a potentially life threatening illness and outbreak on only the visual inspection of the skin lesions. The author's reference the WHO case surveillance definition for suspected MPXV infection which includes the presence of typical skin lesions and the low likelihood of an alternative diagnosis (varicella zoster, herpes zoster, measles, herpes simplex, bacterial skin infections, disseminated gonococcus infection, primary or secondary syphilis, chancroid, lymphogranuloma venereum, granuloma inguinale, molluscum contagiosum, allergic reaction (e.g., to plants); and any other locally relevant common causes of papular or vesicular rash). The authors state that given the severity of Clade I MPVX with high rates of transmission and mortality, it is important not to rely on visual assessment of the skin lesions and it is very important to increase access to specific diagnostics (currently nucleic acid testing) in central Africa where Clade I is transmitted

This study regarding Clade I MPVX infection compliments the author's prior studies regarding the more common Clade II MPVX infection.

Reviewer #2: (No Response)

**Editorial and Data Presentation Modifications?**

Reviewer #1: Several of the figures seem of limited value to me, without adding to the information in the narrative and tables, if the number of figures is an issue for the publisher. There needs to be more detail about the makeup of the persons who were sent the survey, not just a table of where respondents were based and self report on expertise and number of patients treated. There should be an acknowledgement that clinicians have more data than just the appearance of the lesions when making a diagnosis. The WHO definition is not only based on the appearance of the lesions, as the authors suggest, but also on information that help judge the likelihood of alternate diagnoses. 

With these clarifications, the paper could be accepted.

Reviewer #2: (No Response)

**Summary and General Comments**

Reviewer #1: It is a cleverly managed study regarding an illness of rapidly increasing importance. Although there is widespread agreement that access to pathogen specific diagnostics for Clade I MPVX in remote parts of central Africa (DRC, CAR, Cameroon, Sudan, and South Sudan), this study adds to the growing knowledge of this very important illness and pathogen.

Reviewer #2: I commend the authors for this study, which highlights the critical issue of diagnostic capacity in developing countries, often resulting in incorrect therapeutic interventions. However, it's important to acknowledge that diagnosis is multifaceted, relying not only on images but also on factors like risk assessment, patient history, and physical examination. While the study emphasizes the role of images, it should discuss the broader diagnostic process in its introduction and discussion. Additionally, exploring prior research on the role of images in diagnosing diseases like mumps, smallpox, and varicella, which primarily present with characteristic rashes, would enhance the study's context and significance.

PLOS authors have the option to publish the peer review history of their article (what does this mean?). If published, this will include your full peer review and any attached files.

Reviewer #1: Yes: Jonathan Allen Cohn, MD MS FACP FIDSA

Reviewer #2: Yes: Dawd Siraj

Figure Files:

Data Requirements:

Reproducibility:

References

---

## [Editor Report · Decision Letter 1]

4 Jun 2024

Dear Ms Bourner,

We are pleased to inform you that your manuscript 'Challenges in Clinical Diagnosis of Clade I Mpox: Highlighting the Need for Enhanced Diagnostic Approaches' has been provisionally accepted for publication in PLOS Neglected Tropical Diseases.

Best regards,

Wen-Ping Guo

Academic Editor

Michael Holbrook

Section Editor

---

## [Editor Report · Acceptance letter]

17 Jun 2024

Dear Ms Bourner,

We are delighted to inform you that your manuscript, "Challenges in Clinical Diagnosis of Clade I Mpox: Highlighting the Need for Enhanced Diagnostic Approaches," has been formally accepted for publication in PLOS Neglected Tropical Diseases.

Best regards,

Shaden Kamhawi

co-Editor-in-Chief

Paul Brindley

co-Editor-in-Chief
